# LEARNING REPRESENTATIONS THROUGH CONTRASTIVE NEURAL MODEL CHECKING

## ABSTRACT

Model checking is a key technique for verifying safety-critical systems against formal specifications, where recent applications of deep learning have shown promise. However, while ubiquitous for vision and language domains, representation learning remains underexplored in formal verification. We introduce Contrastive Neural Model Checking (CNML), a novel method that leverages the model checking task as a guiding signal for learning aligned representations. CNML jointly embeds logical specifications and systems into a shared latent space through a self-supervised contrastive objective. On industry-inspired retrieval tasks, CNML considerably outperforms both algorithmic and neural baselines in cross-modal and intra-modal settings. We further show that the learned representations effectively transfer to downstream tasks and generalize to more complex formulas. These findings demonstrate that model checking can serve as an objective for learning representations for formal languages.

## 1 INTRODUCTION

Design errors or flaws, particularly in hardware or safety-critical systems, can result in large financial and reputational damage (Baier & Katoen, 2008). To combat this, formal verification methods are deeply integrated into most modern Electronic Design Automation (EDA) tools and are used by many major software and hardware design companies. One of the main verification paradigms for proving system properties is model checking. It has been used to verify drivers, communication protocols, real-time systems, and many other applications (Clarke et al., 2018), and its impact has been recognized in academia and industry (Clarke et al., 2009).

However, despite the research and advancements in the field (Clarke & Wang, 2014), limitations such as the state space explosion problem (Clarke et al., 2011) complicate usage of model checking for many real-world scenarios. Concurrently, deep learning has achieved remarkable results in related fields of Boolean Satisfiability (SAT) (Selsam & Bjørner, 2019; Selsam et al., 2019) and theorem proving (Han et al., 2021; Bansal et al., 2019; Paliwal et al., 2020). This has motivated early applications of deep learning to model checking (Giacobbe et al., 2024; Zhu et al., 2019; Xu & Lieberherr, 2022) as well as to other verification tasks (Wu et al., 2024; Luo et al., 2022).

Most of the existing work on deep learning for verification has focused on learning formal tasks, with far less focus being spent on the need for aligned representations. Verification procedures such as model-checking typically involve two distinct formal languages for describing the system and the specification (Baier & Katoen, 2008). While feature engineering methods have shown success when working with a single formal language (Kretínský et al., 2025; Lu et al., 2025), aligning representations over two modalities brings additional challenges to an already difficult domain.

In this paper, we present a novel method for learning aligned representations of formal semantics by using the model checking task as a contrastive learning objective for a bi-encoder model. We present a self-supervised learning approach, which combined with a scalable technique for generating large datasets, enables the *Contrastive Neural Model Checking (CNML)* model to learn aligned representation of two semantics jointly used for verification, aligned in a shared latent space. We demonstrate our method on learning two important semantics used in verification: specifications expressed as formulas in Linear Temporal Logic (LTL) (Pnueli, 1977) and systems represented as sequential circuits in the AIGER format (Brummayer et al., 2007).

The architecture and the training objective efficiently use the available dataset, avoiding expensive computations needed for a fully supervised approach. The proposed architecture is agnostic to the syntax of specifications and systems, which allows for easy transfer to different logics or circuit encodings, and removes the need for specialized transformer architectures.

We evaluate on example tasks motivated by industry practices, showing high $Recall@1\%$ and $Recall@10\%$ for both cross-modal and intra-modal tasks, outperforming both algorithmic and neural baselines on all metrics. Furthermore, the utility of the learned representations is demonstrated for downstream finetuning for related tasks, with CNML successfully learning transferable representations. We show that our approach leads to a model that can generalize from simple formulas. We further show that the learned embeddings carry information beyond the samples seen in training data, and that the model can learn complex semantic concepts without explicit supervision.

In this work, we make the following contributions:

1. We introduce a joint-embedding model architecture based on the model checking task for AIGER circuits and LTL specifications, which learns aligned embeddings through a self-supervised contrastive approach. We present a simple and efficient method to generate, and also augment, model checking datasets.

2. We demonstrate the ability of the model to learn semantics of both circuits and specification, and to learn both cross-modal and intra-modal relationships. We show that our model can be used for tasks such as retrieval via similarity search. Furthermore, we show that the representations can transfer to downstream tasks.

3. We show that representations learned on simple specifications generalize to complex formulas and transfer effectively to downstream tasks. This shows that by appropriately structuring our learning objective, we can successfully learn aligned representations and the underlying semantics.

## 2 RELATED WORK

Deep Learning has proven itself useful in working with formal logics (Li et al., 2024), with success in both automated (Bansal et al., 2019; Paliwal et al., 2020) and interactive theorem proving (Mikula et al., 2024; Han et al., 2021), Boolean Satisfiability (SAT) (Selsam & Bjørner, 2019; Selsam et al., 2019; Ghanem et al., 2024) and Satisfiability Modulo Theories (SMT) (Balunovic et al., 2018). Mikula et al. (2024) in particular effectively use contrastive learning for premise selection in theorem proving. Our work differs from this general direction by focusing on temporal logics, which are particularly important in verification, and by working on developing aligned representations of different semantics - something not explored in the wider field of machine learning for logics.

In particular, machine learning has been applied in the domain of Linear-Time Temporal Logic (LTL). Most of the existing work has focused on traces (Camacho & McIlraith, 2019; Neider & Gavran, 2018; Walke et al., 2021; Luo et al., 2022). A transformer-based approach in Hahn et al. (2021) shows both the ability of neural generation of propositional assignments and, importantly, the ability of transformers to generalize to LTL. Recent work by Kretínský et al. (2025) uses hand-crafted features of LTL derived game-arenas to guide an algorithm for synthesis. In contrast to these works, we focus on learning representations of LTL formulas, rather than on particular tasks related to traces or assignments.

Due to the wide usage of AIGER in industry, there has been a large variety of work on developing methods for learning the representation of circuits, ranging from GNNs to LLMs (Shi et al., 2024; Zheng et al., 2025; Zhu et al., 2022). Recent works by Wu et al. (2025) and Fang et al. (2025) are based on learning representations of circuits in alignment with properties of hardware description language and hardware circuit code to enable specific tasks in the hardware domain. However, there has been limited work in learning representations aligned to formal specifications, with the closest being by Lu et al. (2025) which uses graph kernel methods to extract features from circuits and select the optimal verification algorithm for the instance.

Machine learning research combining circuits and specifications has primarily concentrated on neural circuit synthesis and neural model checking. Schmitt et al. (2021) propose a neural approach for reactive synthesis (Church, 1963) using hierarchical transformers, while Cosler et al. (2023)

demonstrate that transformers can perform circuit repair against a formal specification. Most recently, Giacobbe et al. (2024) obtain sound neural model checking by learning ranking functions, but their method is targeted at solving individual problem instances. Other approaches recast model checking in different paradigms: Xu & Lieberherr (2022) frame it as a run-time problem solved with Monte Carlo Tree Search, while Madusanka et al. (2023) treat it as a natural-language-style task. Prior work on circuits and specifications has concentrated on learning direct tasks. Our work is primarily concerned with using neural model checking as a proxy to learn aligned representations of both circuits and specifications.

## 3 BACKGROUND

**Linear-Time Temporal Logic (LTL).** Linear-Time Temporal logic (LTL) (Pnueli, 1977) is widely adopted in both academic and industrial settings (Baier & Katoen, 2008). It serves as the foundation for hardware specification languages like Property Specification Language (PSL) (IEEE-Commission, 2005) and System-Verilog Assertions (SVA) (IEEE-Commission, 2024) used in industry.

LTL combines propositional boolean logic operators such as $\neg, \wedge, \vee, \rightarrow$ with temporal operators such as $\bigcirc$ - *next*, $\mathcal{U}$ - *until*, $\square$ - *always*. Temporal operators enable reasoning about sequences of events. As an example, the following simple formula describes that as long as $i_0$ is true, whenever $i_1$ does not hold, in the next step $o1$ should be true.

$$\varphi = (\square\, i_0) \rightarrow (\square\, (\neg i_1 \rightarrow \bigcirc o_1))$$

As LTL does not have a standard normal form, we work with the *assume-guarantee* format as our de-facto normal form. This format syntactically separates assumptions from guarantees, both composed of conjunctions of LTL sub-formulas. Guarantees describe behaviors that we want to verify in our system and assumptions describe the situations in which guarantee properties have to hold. The format is generally given in the form of

$$spec := (assumption_1 \wedge \ldots \wedge assumption_n) \rightarrow (guarantee_1 \wedge \ldots \wedge guarantee_m)$$

We provide a complete definition of LTL syntax and semantics in Appendix A.

**And-Inverter Graphs.** In this paper, we represent sequential circuits as And-Inverter Graphs. And-Inverter Graphs, and particularly the ASCII-encoded AIGER (Brummayer et al., 2007), allow for a succinct representation of hardware circuits in text form and are widely used in both academia and industry. Circuits are built by connecting input variables to output variables through connections of logical gates (AND-Gate and NOT-Gate) and memory cells (latches). For a simple example of an AND-Inverter Graph and its AIGER representation, see Figure 1. We fully define the AIGER format in Appendix B.

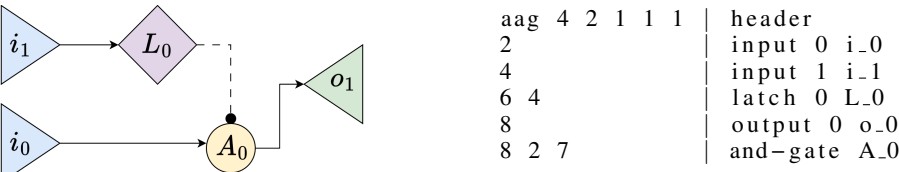

Figure 1: Visualization of a simple circuit represented as an And-Inverter Graph and the corresponding AIGER text representation. The circuit models the behavior described by the formula $\varphi$.

**Model Checking.** Formally, model checking is an automated way of determining whether a model of a system $S$ satisfies a given formal specification $\varphi$ of some desired behavior (Clarke et al., 2018). The desired behavior is formalized into a specification through some logic such as LTL, CTL, PSL, or others. Systems are commonly modeled using circuits or transition systems. A system satisfies some property if and only if the specification holds for the output of the circuit for all possible input traces. We denote it as $S \models \varphi$ (system $S$ satisfies the property $\varphi$).

Model checking algorithms, in general, have three possible outcomes (Baier & Katoen, 2008). The first possible outcome is a result that our specification holds on our model, meaning that the model

*satisfies* the specification. The second possible outcome is that the model *violates* the specification, in which case the algorithm generates a witness for the behavior of the circuit which violates the specification. The third outcome is that the model checking algorithms run out of time and/or memory, which happens when the state space of a problem is too large to be handled algorithmically.

**Contrastive Learning.** The main idea of contrastive learning is that models should also learn from negative samples, not just the positive ones. Contrastive learning enables the development of more robust (Xue et al., 2022) and discriminative representations (Le-Khac et al., 2020). The technique's great success in Computer Vision (Chen et al., 2020; Radford et al., 2021; Khosla et al., 2020) motivated its spread into Natural Language Processing, where it has achieved many strong results (Wu et al., 2020; Ho & Vasconcelos, 2020; Chen et al., 2020). It has demonstrated capabilities in zero-shot learning (Rethmeier & Augenstein, 2023), resilience to noisy datasets (Jia et al., 2021), efficacy in transfer learning (Radford et al., 2021), good performance on semantic textual similarity tasks (Gao et al., 2021), and generalization to unseen inputs (Pappas & Henderson, 2019) - as well as initial use in the logic domain (Mikula et al., 2024; Han et al., 2021).

## 4 DATASET

A key driver of the success of modern deep learning, and transformer-based models in particular, is the sheer scale of training data (Kaplan et al., 2020). As large datasets of circuit designs are the intellectual property of hardware design firms, they are typically kept confidential. Unlike in Natural Language Processing, or Computer Vision where data could be scrapped from the internet, there are no large circuit-specification datasets available.

As a consequence, we have to synthetically generate a large, high-quality dataset of satisfying pairs. However, synthetic data generation is challenging due to the high complexity of the verification problem, structure of formal language syntax and semantics, and the need for variety in circuit and specification samples.

Due to the complexity of the underlying semantics, using purely probabilistic approaches for formula generation leads to the generation of syntactically valid formulas that, however, often do not specify interesting behaviors. To address this, we follow the LTL formula generation technique from Schmitt et al. (2021) to generate a diverse set of LTL formulas. Unlike the works of Schmitt et al. (2021) or Cosler et al. (2023), which use the assumptions and guarantees as separate inputs to their hierarchical transformers, we generate specifications by merging all assumptions and guarantees into a single LTL formula.

Generation of corresponding circuits is another significant obstacle, as stochastic methods are unlikely to generate satisfying circuits without a very high number of attempts. Therefore, we have to generate circuits that inherently satisfy the specification formulas. We use reactive synthesis (Church, 1963) to automatically generate satisfying circuits based on each specification. We utilize existing approaches and the Strix LTL synthesis tool (Meyer et al., 2018) to create a diverse dataset of satisfying circuits.

To prevent overfitting on syntactic patterns, we perform several augmentations to the data format. We shuffle the order of assumption LTL formulas for each specification formula, and we enforce a uniform number of input and output wires for all circuits, even if they are not explicitly used. Enforcing a fixed number of input and output wires for every circuit eliminates a "wire-counting" trick that the model could exploit. By standardizing every circuit to the same number of wires, we remove that correlation.

We call the resulting dataset with $295,665$ samples `cnml-base`.

## 5 LEARNING REPRESENTATIONS

The complexity of verification problems (Stockmeyer, 1974; Sistla & Clarke, 1985) presents a significant barrier not only to synthetic data generation, but also to learning. As the underlying symbolic tasks are highly complex, machine learning models tend to prioritize superficial syntactic patterns rather than dealing with the fundamental goal of building semantic understanding.

Furthermore, many verification tasks such as model checking, are inherently bimodal - one formal language talks about the specification (what we want the system to do) while the second one talks about the system model (what the system actually does). While both languages come with their own syntax and semantics, they fundamentally describe the same object. This further complicates training as the learned representations have to encode not just the properties of their own modality, but also the relation to the other one.

## 5.1 MODEL ARCHITECTURE

While supervised learning could be used to learn the semantics of verification based on labels derived from model checking circuit-specification pairs, this is computationally inefficient as it requires all samples to have explicit labels. Furthermore, supervised learning is limited to just one learning signal i.e. the label for a single circuit-specification pair. Circuits are not characterized just by the specification that they satisfy - but also by the specifications that they do not. This observation naturally leads us to contrastive learning, where the learning objective is not defined just by how an input relates to its positive samples, in our case circuits and the specifications that they satisfy, but also by its relationship with the negative samples - the specifications that they violate. Following this idea and inspired by the work of Radford et al. (2021), we adopt a self-supervised contrastive approach for learning aligned representations of circuits and formal specifications.

While Radford et al. (2021) use contrastive learning to align image and text representations, our approach adapts this framework to align representations of circuits and specifications. Our model is trained to project circuit embeddings closer to the embeddings of specifications they satisfy, and farther away from those they do not satisfy. Practically, we view the different semantics and syntaxes of circuits and specifications as different modalities, and learn a joint embedding space for circuit-specification pairs.

Our model uses two distinct text encoders, $E_\varphi$ and $E_c$. Despite the encoders learning over a joint space, $E_c$ and $E_\varphi$ do not share any parameters. While models in related work (Schmitt et al., 2021; Cosler et al., 2023; Radford et al., 2021) are trained from scratch, we initialize both encoders as CodeBERT models (Feng et al., 2020). As shown by Schmitt et al. (2023) for the closely related task of reactive synthesis, pre-trained Transformer models can have a simpler architecture, and achieve similar results.

A single input sample, consisting of a specification and a circuit, is fed into the encoders separately: $E_c$ only sees the AIGER circuit $c$, and $E_\varphi$ sees only the LTL specification $\varphi$. The forward pass through $E_c$ and $E_\varphi$ produces the respective input's sequence embeddings. We take the output of the pooling of their encodings as the intermediate representation of the whole sequence. Both summary vectors are then multiplied by a learned projection matrix (one for $E_c$ and another for $E_\varphi$), which is used to upscale the embedding dimension to 1024.

The use of two independent encoders forces each one to focus on its own modality. This separation prevents overfitting to syntactic patterns that may arise from specific circuit-specification pairings. Additionally, the self-supervised approach enables the implicit construction of negative samples without requiring explicit model-checking of all possible circuit-specification pairs, which would otherwise be computationally infeasible. This allows for generation of a larger corpora, which is easier to augment and does not require manual generation of negative samples.

## 5.2 TRAINING

At the start of each epoch, we construct the mini-batches using a greedy algorithm. The mini-batches are optimized to ensure that they do not contain any duplicate circuits or any duplicate specifications. Furthermore, the algorithm cross-checks off-diagonal samples with the rest of the dataset to minimize the rate of false negatives which we find to be roughly $4\%$.

Based on $N$ circuit-specification pairs $(c_1, \varphi_1), \ldots, (c_N, \varphi_N)$ that are directly known to be positive ($c_i$ satisfies $\varphi_i$), we compute the embeddings of circuits and specifications as described previously, creating embeddings $u_{c_1}, \cdots, u_{c_N}$ and $v_{\varphi_1}, \cdots, v_{\varphi_N}$. We then create all pairwise combinations of circuit embeddings and specification embeddings $(u_{c_i}, v_{\varphi_j}), 0 < i, j \leq N$ through a $N \times N$ matrix. Following that, we calculate the cosine similarity for all pairings by computing a dot product between all the L2 normalized circuit embeddings and the specification embeddings. On the diagonal of the

resulting matrix lie the $N$ embeddings of circuit-specification pairs $(c_1, \varphi_1), \ldots, (c_N, \varphi_N)$ that are directly known to be positive. The remaining $N^2 - N$ pairs $(c_i, \varphi_j)$, where $i \neq j$ and $0 < i, j \leq N$, are implicitly coded negative.

The full training objective consists of two components - where $\mathcal{L}_{\mathrm{CE}}$ is the contrastive component, and $L_{\mathrm{RR}}$ is the regularization component, with $\lambda$ being the weighting factor.

$$\mathcal{L}_{\mathrm{CNML}} = \mathcal{L}_{\mathrm{CE}} + \lambda \, L_{\mathrm{RR}},$$

The contrastive loss is calculated using a symmetric cross-entropy loss function computed over rows and columns of the matrix of similarity scores, following the method from van den Oord et al. (2018). We further augment the contrastive loss with a weighted representation similarity regularization loss, as introduced in Shi et al. (2023). We find that it provides stability during the training, prevents overfitting, and importantly, allows the use of a higher learning rate, without risking catastrophic forgetting common in BERT models (Sun et al., 2019; McCloskey & Cohen, 1989). The forward pass and loss computation are visualized in Figure 2. We report the hyperparameters and the detailed training setup in Appendix C.

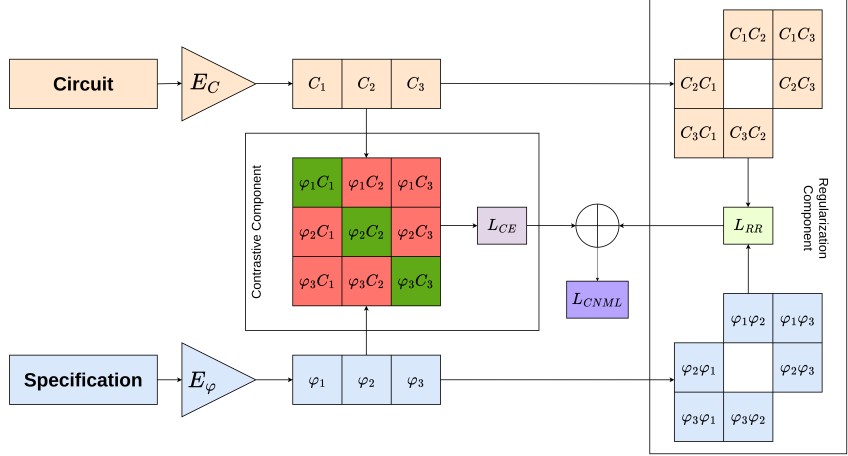

Figure 2: Visualization of the forward pass and the computation of the two loss components.

## 6   MODEL EVALUATIONS

We train two models: CNML-base trained on the `cnml-base` dataset for demonstrating the performance of our method on various tasks, and CNML-simple trained on the `cnml-split` dataset of simple formulas, designed to showcase the model's ability to generalize (described in detail in Section 6.4). We evaluate the learned embeddings by inspecting the latent space and distribution of cosine similarity scores between various circuit-specification pairs, and by assessing performance on two retrieval tasks based on real-world problems from Computer-Aided Design, as well as downstream fine-tuning for the model checking task.

### 6.1   EMBEDDING SPACE ANALYSIS

We inspect the learned embedding space by observing the distributions of the cosine similarity that our model produces on the test split of `cnml-base`. For a dataset-level insight, Figure 3a plots the distributions of cosine similarity values that the model attributes to positive (circuit satisfies the specification) and negative (circuit violates the specification) pairs. Both distributions are normalized to the probability density function, with the red distribution showing negative, and the green positive circuit-specification pairs. For a batch level insight, Figure 3b shows a normalized heatmap of the similarity matrix for a singular batch from the test dataset.

Both visualizations in Figure 3 show that the model is able to separate satisfying from violating pairs of circuits and specification. Figure 3a shows that the model effectively separates the two

distributions, with a small remaining overlap. On the heatmap plot, we see that the model produces the highest cosine similarity values on the diagonal - the satisfying pairs of circuits and specifications.

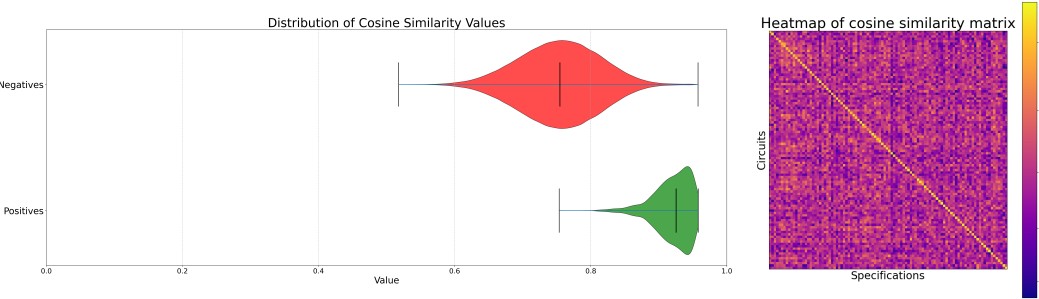

(a) Distributions of cosine similarity values for the positive (green) and negative (red) circuit-specification pairs

(b) Cosine similarity heatmap

Figure 3: Visualization of the Cosine Similarity Distribution produced by the CNML-base model

## 6.2 RETRIEVAL

We evaluate our model on two retrieval tasks. The first task is *cross-modal* retrieval: given an LTL specification, we seek to retrieve a matching design from a collection of candidate circuits. By retrieving an existing design, it is possible to avoid the computational expense of automatic synthesis or the effort of manual design. Archiving and reusing existing circuits is a common occurrence in industry and is supported by many commercial tools (Fang et al., 2025). The second evaluation task is an *intra-modal* retrieval task, in which we look for potential optimization replacements for a given circuit. Even when an automated tool or an engineer generates a circuit that satisfies the formal specification, the result may lack desirable properties such as minimal gate count, wire placement, or manufacturability.

We generate two test retrieval datasets through mining the test split of `cnml-base`. The first dataset consists of 127 test batches, each of size $N = 100$, where exactly one circuit is a matching candidate while all others do not satisfy the main specification. In the same way, we construct the second dataset with 100 test batches of size $N = 1000$.

We compare the CNML models against several baseline methods. Bag-of-Keywords and Weisfeiler-Lehman Graph Kernels (Shervashidze et al., 2011) were recently used for feature extraction of circuits in Lu et al. (2025). For a text-edit based similarity metric, we use the Inverted Levenshtein distance. For machine-learning baselines we compare against the CodeBert model without any CNML pre-training, and against a bi-encoder model following the Sentence-BERT architecture (Reimers & Gurevych, 2019), to which we refer as Siamese-CNML.

We measure Mean Reciprocal Rank (MRR), Mean Rank (MR) and the $Recall@1\%$ ($R@1\%$) and $Recall@10\%$ ($R@10\%$) values which measure the recall metric for the top $1\%$ and $10\%$ of the batch, respectively. We report the results for cross-modal retrieval in Table 1, and for intra-modal in Table 2.

Table 1: Cross-modal Results for Different Methods and Dataset Sizes.

| Method | $127 \times$ N=100 | | | | $100 \times$ N=1000 | | | |
|---|---|---|---|---|---|---|---|---|
| | MRR | MR | R@1% | R@10% | MRR | MR | R@1% | R@10% |
| CodeBERT | 0.060 | 46.1 | 0.8% | 4.7% | 0.014 | 471.65 | 2.0% | 9.0% |
| Siamese-CNML | 0.043 | 49.7 | 0.0% | 9.4% | 0.005 | 467.53 | 1.0% | 7.0% |
| CNML-simple | 0.118 | 34.5 | 4.7% | 24.4% | 0.099 | 286.12 | 15.0% | 39.0% |
| **CNML-base** | **0.286** | **16.6** | **16.5%** | **61.4%** | **0.195** | **188.03** | **38.0%** | **62.0%** |

Table 2: Intra-modal Results for Different Methods and Dataset Sizes.

| Method | $127 \times$ N=100 | | | | $100 \times$ N=1000 | | | |
|---|---|---|---|---|---|---|---|---|
| | MRR | MR | R@1% | R@10% | MRR | MR | R@1% | R@10% |
| Inverted Levenshtein | 0.068 | 49.2 | 3.1% | 11.0% | 0.038 | 472.4 | 5.0% | 12.0% |
| Bag-of-keywords | 0.055 | 44.2 | 0.7% | 12.6% | 0.014 | 456.1 | 0.0% | 11.0% |
| Weisfeiler–Lehman | 0.066 | 47.6 | 2.4% | 12.6% | 0.023 | 469.3 | 7.0% | 13.0% |
| CodeBERT | 0.054 | 50.2 | 1.6% | 9.5% | 0.029 | 468.4 | 4.0% | 12.0% |
| Siamese-CNML | 0.056 | 48.7 | 1.6% | 8.7% | 0.021 | 456.9 | 4.0% | 16.0% |
| CNML-simple | 0.190 | 25.9 | 11.0% | 35.4% | 0.124 | 229.6 | 21.0% | 52.0% |
| **CNML-base** | **0.252** | **18.9** | **13.4%** | **52.7%** | **0.164** | **199.8** | **31.0%** | **58.0%** |

Results in both tables show that the CNML-base model significantly outperforms all baseline methods across both scenario sizes. The advantage of CNML-base expands on the larger problem sizes, with an approximately 75% Mean Rank improvement versus the best algorithmic baseline (Weisfeiler-Lehman at 417.5). Overall, these results indicate that CNML representations can capture relevant semantics more effectively than other machine learning or algorithmic approaches, and that they can be used for tasks directly on the embeddings.

## 6.3 DOWNSTREAM FINE-TUNING

We further evaluate CNML as a pre-training objective for downstream fine-tuning. We train models to perform binary classification on circuit-specification pairs to determine whether the circuit satisfies the specification - the model checking task. The architecture follows the Sentence-Bert architecture (Reimers & Gurevych, 2019), with the bi-encoders being followed by a linear probe. The dataset comprises 96940 training examples and 12262 test examples. Models are initialized either from CodeBERT or from our CNML pre-trained encoders, then fine-tuned on the downstream task.

Table 3: Fine-tuning performance on circuit-specification model checking task

| Model | Accuracy | Precision | Recall | F1 Score |
|---|---|---|---|---|
| CodeBERT | 0.830 | 0.799 | 0.884 | 0.839 |
| CNML-simple | 0.845 | 0.814 | 0.894 | 0.852 |
| CNML-base | **0.887** | **0.847** | **0.947** | **0.894** |

The results in Table 3 demonstrate that CNML pre-training provides substantial benefits for downstream performance over the baseline model, where we initialize models with CodeBERT weights and no CNML pretraining. The performance gain over the baseline CodeBert models, shows that the contrastive pre-training objective successfully learns transferable representations that capture the semantic relationship between specifications and circuits.

## 6.4 GENERALIZATION

We set-up an experiment to test the generalization capabilities of our approach. We evaluate the generalization capability of CNML models by training on simple formulas and testing on more complex specifications. To construct a suitable training dataset of circuit-specification pairs, we employ *formula splitting*. This technique allows us to soundly transform the `cnml-base` dataset into one with simpler LTL formulas, while preserving the soundness of circuit-specifications pairs.

Formula splitting systematically weakens specification guarantees to create new formulas. Consider an LTL specification $\varphi$ defined as:

$$\varphi := \bigwedge_{assumption \in \varphi_A} assumption \rightarrow \bigwedge_{guarantee \in \varphi_G} guarantee$$

where $\varphi_A$ and $\varphi_G$ are sets of assumption and guarantee formulas, respectively. For any circuit $\mathcal{C}$ satisfying $\mathcal{C} \models \varphi$ and any guarantee $\varphi' \in \varphi_G$, the following holds:

$$\mathcal{C} \models \bigwedge_{assumption \in \varphi_A} assumption \to \varphi'$$

We use this observation, and apply formula splitting to specifications in `cnml-base` while preserving the original circuit. By doing this, we generate the `cnml-split` dataset and transform the original formulas into ones which contain exactly one guarantee. We train CNML-simple model on this dataset, exposing the model only to single-guarantee formulas during training, while evaluating on multi-guarantee formulas by using `cnml-base` on the same experiments as with CNML-base.

We evaluate the CNML-simple model on retrieval and fine-tuning tasks. Tables 1 and 2 present the performance of CNML-simple on retrieval problems based on specifications more complex than the ones seen during training. The model outperforms all baseline methods on both retrieval tasks, although performance decreases compared to CNML-base due to the distribution shift and the mini-batch noise. Additionally, as shown by fine-tuning results on the model checking task (Section 6.3) reported in Table 3, the learned representations transfer to downstream tasks even when they involve complex formulas.

These results demonstrate that CNML models can generalize from simple training formulas to complex multi-guarantee specifications. Since CNML-simple is exposed to only single-guarantee formulas during training, its successful performance on multi-guarantee test formulas indicates the ability of CNML models to generalize.

## 7 CONCLUSION

In this paper, we introduced CNML, a neural model checking framework that learns joint embeddings of LTL specifications and AIGER circuits. The contrastive self-supervised training approach allows for training using only the positive circuit-specification pairs, and can effectively use such samples to learn aligned representations of both semantics. We create a large dataset of $295, 665$ samples and present a method for data generation and augmentation at scale, potentially enabling future work in machine learning for formal logics and verification - a domain where data is usually scarce and computation is prohibitively expensive.

Evaluation on industry-inspired retrieval tasks shows that CNML outperforms the baselines, achieving high $Recall@1\%$ and $Recall@10\%$ for both cross-modal and intra-modal tasks. We further demonstrate that the learned representations can be used for fine-tuning on downstream task. We show that the method is able to generalize from training on simple formulas, to performing tasks on formulas in more complex formats. Our results validate the effectiveness of self-supervised contrastive pre-training in learning semantics for used in verification.

We believe that the model training paradigm and data generation opens a window for further research where representation alignment of formal languages is crucial. The method presented enables learning of aligned representation - allowing for future work combining formal methods and deep learning in problems such as verification, synthesis and retrieval.

## 8 REPRODUCIBILITY STATEMENT

We provide detailed information required to reproduce our results. Detailed hyperparameters, training set-up and package versions are listed in Appendix C. The dataset generation procedure is described in Section 4, while the model architecture and forward pass are detailed in Sections 5.1 and 5.2, respectively. We will open-source our implementations, trained model checkpoints, and the dataset following the review process.

## 9 LLM'S USE STATEMENT

During the writing of this paper, Large Language Models were used for the purposes of grammar checking and correction, as well as to help improve clarity by suggesting text refinements. They were also used to assist implementation work.

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

## A  LINEAR TEMPORAL LOGIC

Formally, LTL syntax is defined as:

$$\varphi := p \mid \varphi \wedge \varphi \mid \neg\varphi \mid \bigcirc\varphi \mid \varphi\,\mathcal{U}\,\varphi$$

We evaluate LTL semantics over a set of traces: $TR := (2^{AP})^\omega$. For a trace $\pi \in TR$, we denote $\pi[0]$ as the starting element of a trace $\pi$, and for a $k \in \mathbb{N}$, let $\pi[k]$ be the $k$-th element of the trace $\pi$. With $\pi[k, \infty]$ we denote the infinite suffix of $\pi$ starting at $k$. We write $\pi \models \varphi$ for the trace $\pi$ that satisfies the formula $\varphi$.

For a trace $\pi \in TR$, $p \in AP$, and formulas $\varphi$:

- $\pi \models \neg\varphi$ iff $\pi \not\models \varphi$
- $\pi \models p$ iff $p \in \pi[0]$; $\pi \models \neg p$ iff $p \notin \pi[0]$
- $\pi \models \varphi_1 \wedge \varphi_2$ iff $\pi \models \varphi_1$ and $\pi \models \varphi_2$
- $\pi \models \bigcirc\varphi$ iff $\pi[1, \infty] \models \varphi$
- $\pi \models \varphi_1\,\mathcal{U}\,\varphi_2$ iff $\exists l \in \mathbb{N} : (\pi[l, \infty] \models \varphi_2 \wedge \forall m \in [0, l-1] : \pi[m, \infty] \models \varphi_1)$

We further derive several useful temporal and boolean operators. These include $\vee$, $\implies$, $\Leftrightarrow$ as boolean operators and the following temporal operators:

- $\varphi_1\,\mathcal{R}\,\varphi_2$ (release) is defined as $\neg(\neg\varphi_1\,\mathcal{U}\,\neg\varphi_2)$
- $\square\varphi$ (globally) is defined as $\bot\,\mathcal{R}\,\varphi$
- $\Diamond\varphi$ (eventually) is defined as $\top\,\mathcal{U}\,\varphi$ -

# B   AIGER

The format is based on using And-Inverter graphs to concisely describe circuits composed of AND and NOT gates, as well as simple memory cells called latches. More complex circuits are built by combining these elementary components through circuit connections. We represent these connections between inputs, outputs, gates and latches through integer-denoted variables.

- Each circuit variable is represented by a pair of consecutive integers. Odd integers denote the negation of the variable represented by the preceding even integer. The initial variables 0 and 1 represent the constant values FALSE and TRUE, respectively.
- Input and output connections are each defined by a single variable.
- AND gates are specified using three variable numbers. The first variable represents the gate's output, which is the conjunction of the two variables represented by the remaining two numbers.
- Latches function as simple memory cells. Each latch is defined by two variable numbers: the output variable and the input variable. The output variable's value is determined by the input variable's value from the previous computation step. These variables are initially set to FALSE.

The file containing an AIGER circuit begins with a header containing the string `aag` and five numbers (M,I,L,O,A), each representing the size and shape of the circuit.

- **M** : maximum variable index ($2\times$ number of variables)
- **I** : number of inputs
- **L** : number of latches
- **O** : number of outputs
- **A** : number of AND gates

Following the header, each subsequent line represents either an input, latch, output, or gate, adhering to the formatting conventions discussed in this section. After the main body containing the circuit description, there is an optional symbols table which allows for arbitrary naming of all circuit components.

# C   REPRODUCIBILITY

We rely on the PyTorch 2.3.0 (Paszke et al., 2019) and Huggingface Transformers 4.46.2 (Wolf et al., 2019) packages. We train the model on 8 NVIDIA A100-SXM4-80GB GPUs using Distributed Data Parallel and Mixed-precision (Micikevicius et al., 2017).

We train the model for 12 epoch, with a per-GPU mini-batch size of 128 and a gradient accumulation step size of 2. We take the model at 6 epochs as the best one (roughly $110,000$ steps or taking 8 hours of training time ). We use AdamW (Loshchilov & Hutter, 2019) as the optimizer of choice, with the default $\beta_1 = 0.9, \beta_2 = 0.999$ values. The weighing parameter for the representation regularization is set to $\lambda = 0.25$. We initialize the learnable temperature parameter to $\tau = 0.07$ same as in Radford et al. (2021).

The learnable projection matrices are set to project to $1024$ dimensions and are initialized in the same way as in Radford et al. (2021). We find that while going from 768 to 1024 helps the model, there are diminishing returns in increasing dimension higher then that and therefore we keep it at $1024$.

We diverge from Radford et al. (2021) by keeping the logit scaling factor fixed. We use a relatively low value for the learning rate, of $2e^{-4}$, due to the nature of BERT models and the catastrophic forgetting problem appearing at higher values (Sun et al., 2019). We use a linear warm-up and decay scheduler policy with a warm-up period of 4200 steps and a linear decay policy to 0.

Table 4: Hyperparameters and Training Setup

| Category | Hyperparameter | Value / Range |
|---|---|---|
| Hardware & Software | Framework & version
CUDA
GPUs
Random seed | Python 3.10.12,
CUDA 12.3
$8\times$ A100-SXM4-80GB
580946 |
| Training duration | Max epochs
Best checkpoint | 100 ($\approx 165,000$ steps, $\approx 15$ h)
Epoch 80 ($\approx 130000$ steps, $\approx 12$ h) |
| Batching | Per-GPU batch size
Gradient accumulation | 128
2 steps |
| Optimizer & LR | Optimizer
$\beta_1, \beta_2$
Weight decay
Initial LR | AdamW
0.9, 0.999
0.01
$2 \times 10^{-4}$ |
| Scheduler | Warmup steps
Decay policy | 12000
Linear to 0 over $165,000$ steps |
| Model-specific | $\lambda$ (reg. weight)
$\tau$ (temp. init)
Projection dimension | 0.25
0.07
1024 |

## D  R1 - EXECUTION TIME

In Table 5 and Table 6, we provide the wall clock times for the different methods used for the cross-modal and intra-modal retrieval experiments from Section 6. Note that the wall-clock times do not differ significantly between the ML models (CodeBERT, Siamese-CNML, CNML-base), as they have the same architecture and model size for inference.

Table 5: Wall-clock time measurements over methods for intra-modal retrieval

| | $127 \times$ n=100 | | | $100 \times$ N=1000 | | |
|---|---|---|---|---|---|---|
| Wall-Clock Time (s) | Total | Mean | Std | Total | Mean | Std |
| Inverted Levenshtein | 0.032 | 0.001 | 0.001 | 0.314 | 0.003 | 0.001 |
| Bag-of-keywords | 0.101 | 0.001 | 0.002 | 0.660 | 0.001 | 0.002 |
| Weisfeiler–Lehman | 4.232 | 0.033 | 0.002 | 48.520 | 0.485 | 0.055 |
| CodeBERT | 14.509 | 0.114 | 0.007 | 112.951 | 1.129 | 0.021 |
| Siamese-CNML | 14.459 | 0.114 | 0.007 | 113.130 | 1.131 | 0.027 |
| CNML-base | 14.426 | 0.114 | 0.007 | 112.690 | 1.127 | 0.022 |
| *MC-ALL* | 2313.234 | 18.214 | 58.100 | 18213.204 | 182.132 | 448.370 |

Table 6: Wall-clock time measurements over methods for cross-modal retrieval

| | $127 \times$ n=100 | | | $100 \times$ n=1000 | | |
|---|---|---|---|---|---|---|
| Wall-Clock Time (s) | Total | Mean | Std | Total | Mean | Std |
| CodeBert | 14.851 | 0.116 | 0.007 | 115.153 | 1.151 | 0.028 |
| Siamese-CNML | 14.853 | 0.116 | 0.007 | 114.748 | 1.147 | 0.028 |
| CNML-base | 14.847 | 0.116 | 0.007 | 114.672 | 1.146 | 0.029 |
| *MC-ALL* | 2313.234 | 18.214 | 58.100 | 18213.204 | 182.132 | 448.370 |

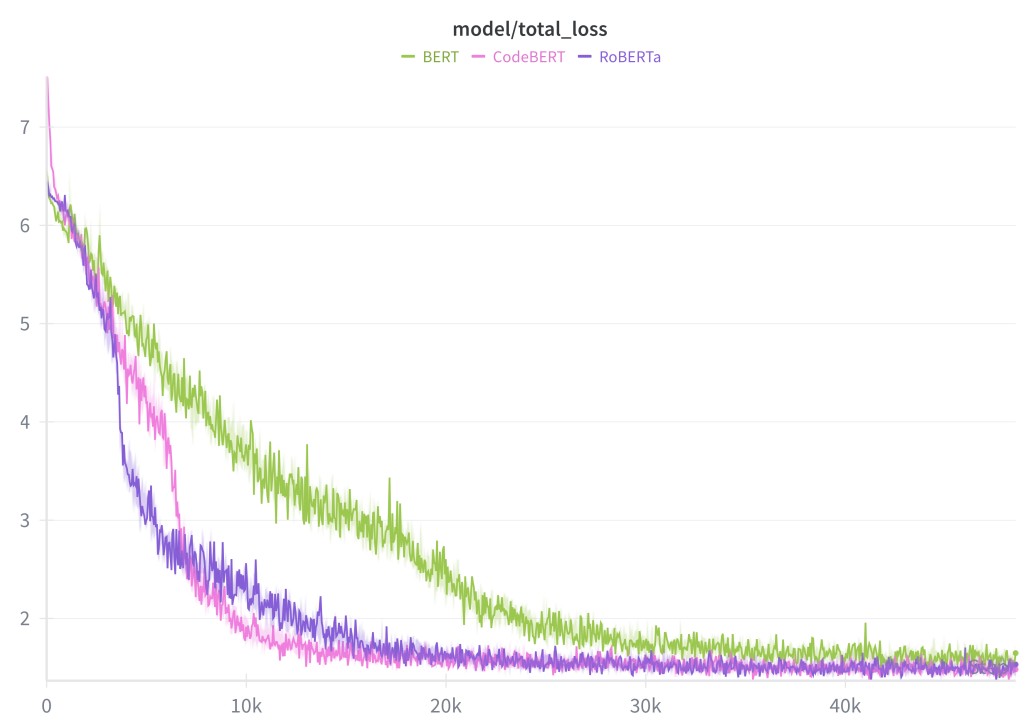

Figure 4: Loss plot of a BERT, RoBERTa, and CodeBERT models

# E    R2 - COMPARISON BETWEEN BERT MODELS

In Figure 4, we compare the performance of BERT (Devlin et al., 2019), RoBERTa (Liu et al., 2019), and CodeBERT (Feng et al., 2020) models with equivalent hyperparameters over the cnml-base dataset. We find that RoBERTa achieves comparable results to CodeBert but takes longer to converge to the same loss, indicating that the code pretraining of CodeBert provides some benefits. BERT does take significantly longer to converge, and training BERT is more sensitive to hyperparameters. Additionally, the comparison with BERT is skewed as it uses WordPiece tokenization (Wu et al., 2016) while RoBERTa and CodeBert use Byte-Pair Encoding (Sennrich et al., 2016) tokenization (Devlin et al., 2019; Liu et al., 2019). Therefore, BERT's token sequences are longer for the same input; hence, the context length of the pre-trained BERT model is de facto shorter.

# F    R3 - ANALYSIS OF RANK DISTRIBUTIONS

In Figure 5 we show histograms of rank distribution and their Cumulative Frequency Distribution (CFD) for all methods for intra-modal search over the $n = 100$ problem set.

We find that both CNML-base and CNML-simple significantly outperform the other methods, with a high concentration of ranks at low values, as demonstrated by the histogram distributions and the CFD curves. Both the neural baselines in Siamese-CNML and CodeBERT, and the algorithmic baselines in Weisfeiler-Lehman, Inverted Levenstein, and Bag-of-keyword, are behind the CNML methods' baselines, with an almost random distribution of ranks. Although we note an interesting property of the Bag-of-keyword method, where, despite its on-average poor performance, it has no ranks in the last quintile, indicating that it does not make 'extreme mistakes' in ranking pairs.

# G    R4 - QUALITATIVE SAMPLES

Figures 6, 7, and 8 present several qualitative samples of circuits and specifications from our dataset. The visualization of circuits, and the computation of the Manna-Pnueli hierarchy, are performed

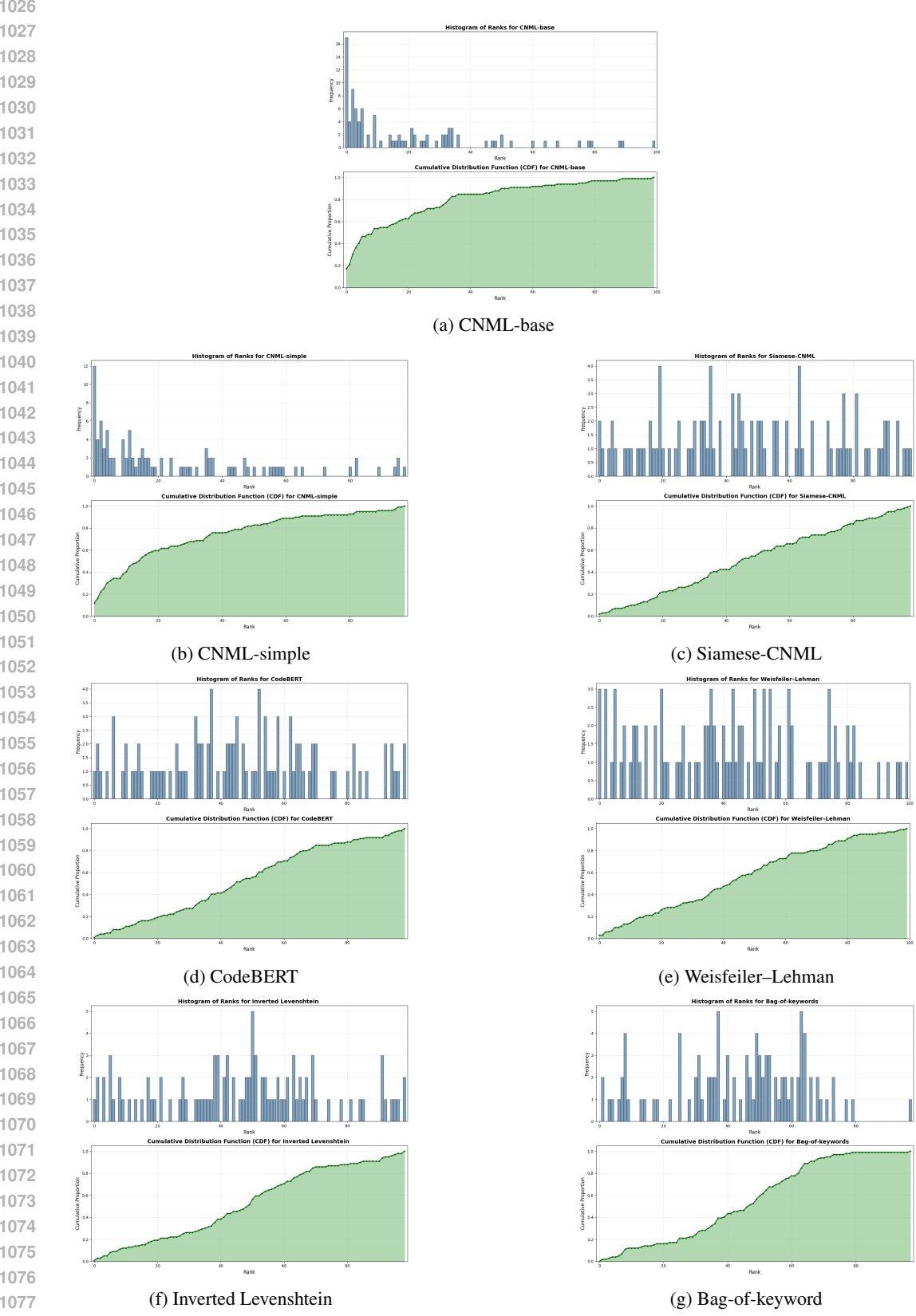

Figure 5: Rank Distribution and CFD plots for all methods

using Spot (Duret-Lutz et al., 2022). Unused input and output gates are removed from the circuit visualization for the purposes of clarity.

The Manna-Pnueli hierarchy (Manna & Pnueli, 1990) classifies specifications in linear-time temporal logic into a hierarchy based on the automata complexity needed to recognize such properties. Its core are safety and guarantee properties. Safety properties describe behavior that can be refuted by a finite counterexample, such as "X never happens". The dual of a safety property is a guarantee property, which describes properties such as "Y happens eventually". The hierarchy is built through the Boolean combination of these classes. The most general class, reactivity, captures all $\omega$-regular properties expressible in LTL. We use this hierarchy as a means to measure the complexity of our specifications, with higher classes representing more intricate specifications.

$$((((\Box(\Diamond(i1))) \vee (\Box(\Diamond(i4)))) \vee (\Box(\Diamond(i0)))) \leftrightarrow (\Box(\Diamond(o1))))$$

(a) LTL Specification

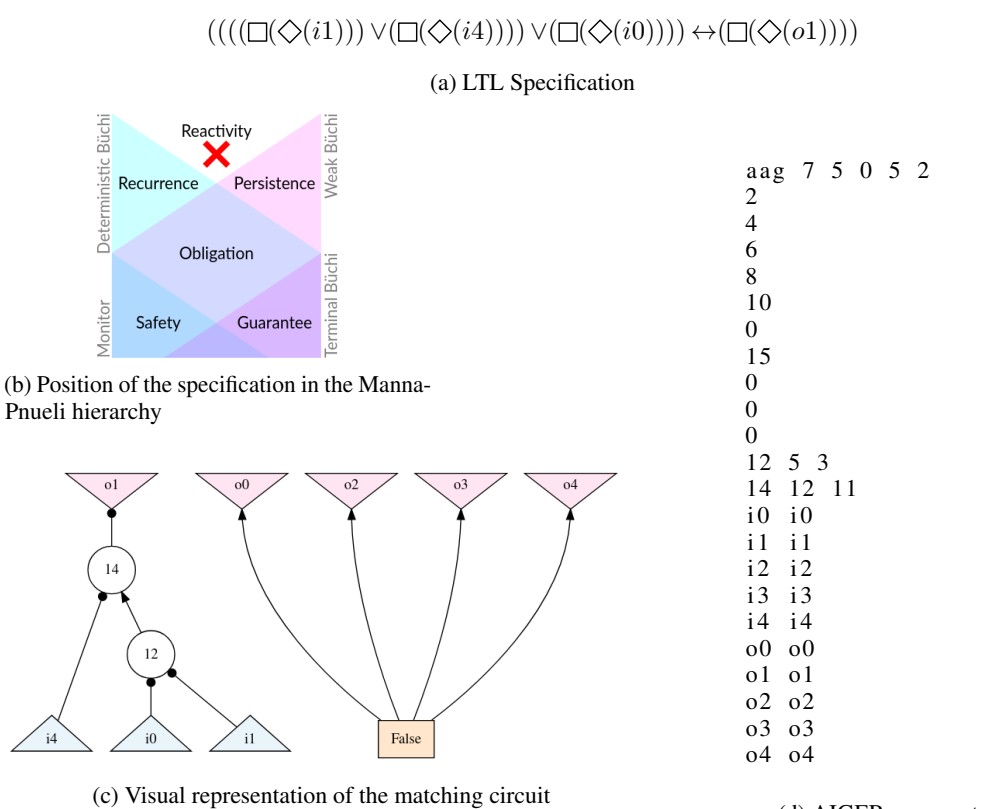

(b) Position of the specification in the Manna-Pnueli hierarchy

(c) Visual representation of the matching circuit

(d) AIGER representation

Figure 6: Example of a circuit-specification pair from cnml-base dataset

## H  R5 - AMBA RESULTS

Table 7: AMBA Cross-modal Results for Different Methods and Dataset Sizes.

|  | 12 samples of N=100 | | | |
| --- | --- | --- | --- | --- |
| Method | MRR | MR | R@1% | R@10% |
| CodeBERT | 0.020 | 54.91 | 0.0% | 0.0% |
| Siamese-CNML | 0.018 | 59.91 | 0.0% | 0.0% |
| CNML-simple | 0.489 | 5.33 | 33.33% | 75.0% |
| **CNML-base** | **0.688** | **1.41** | **58.38%** | **100%** |

$$(\Box(\Diamond((!(i1)) \lor (\bigcirc(o0)))))) \land (\Box(((o1) \land (!(o0))) \leftrightarrow ((!(o0)) \lor (o1)))) \land (\Box((i2) \rightarrow (\Diamond(o1))))$$
$$\land \ (\Box((!(o2)) \lor (!(o4)))) \land (\Box(((((!(i3)) \land (i0)) \land (!(i2))) \land (i1))$$
$$\rightarrow (\Diamond(((((!(o1)) \land (o3)) \land (!(o2))) \land (o4)))))$$

(a) LTL Specification

(b) Position of the specification in the Manna-Pnueli hierarchy

(c) Visual representation of the matching circuit

```
aag 12 5 2 5 5
2
4
6
8
10
12 24
14 22
18
22
0
21
21
16 6 15
18 13 17
20 12 14
22 19 21
24 6 18
i0 i0
i1 i1
i2 i2
i3 i3
i4 i4
l0 l0
l1 l1
o0 o0
o1 o1
o2 o2
o3 o3
o4 o4
```

(d) AIGER representation

Figure 7: Example circuit-specification pair from cnml-base dataset

$(((i1 \wedge i2 \wedge !i3) \, \mathcal{R}(!o2 \vee o3 \vee !o4)) \wedge \square((!o0 \vee !o1) \wedge (!o0 \vee !o3) \wedge (!i1 \vee !o0) \wedge \diamondsuit(!i2 \vee o2)$

$\wedge (!i4 \vee (i0 \wedge \bigcirc(!o0 \wedge !o2 \wedge o3)) \vee (!i0 \wedge \bigcirc(o0 \vee o2 \vee !o3)))$

$\wedge (!o3 \vee \bigcirc(i3 \, \mathcal{R}((!i3 \vee o2) \wedge (i3 \vee o4))))$

$\wedge (!i2 \vee \diamondsuit o3) \wedge (i0 \vee !i1 \vee i2 \vee i3 \vee \diamondsuit(!o1 \wedge !o2 \wedge o3 \wedge !o4)) \wedge (i4 \vee (o4 \wedge \bigcirc o4) \vee (!o4 \wedge \bigcirc !o4))$

$\wedge (!o4 \vee \bigcirc(o2 \, \mathcal{R} \, o1)) \wedge (!i1 \vee \diamondsuit o2)$

$\wedge (!i3 \vee \bigcirc(!i1 \vee \bigcirc\bigcirc\bigcirc\bigcirc\bigcirc\bigcirc\bigcirc(o3 \wedge o4)))))$

$\vee \bigcirc\diamondsuit((o3 \wedge ((i0 \vee i3) \, \mathcal{R}(!i0 \vee i3))) \vee (o2 \wedge ((i0 \vee i2) \, \mathcal{R}(i0 \vee !i2))))$

(a) LTL formula

(b) Position of the specification in the Manna-Pnueli hierarchy

(c) Visual representation of the matching circuit

```
aag 16 5 2 5 9
2
4
6
8
10
12 33
14 21
0
0
24
28
20
16 9 13
18 14 17
20 13 15
22 3 21
24 19 23
26 3 12
28 15 26
30 9 19
32 22 30
i0 i0
i1 i1
i2 i2
i3 i3
i4 i4
l0 l0
l1 l1
o0 o0
o1 o1
o2 o2
o3 o3
o4 o4
```

(d) AIGER representation

Figure 8: Example circuit-specification pair from cnml-base dataset

