# OpenReview forum: "Learning Representations Through Contrastive Neural Model Checking"
_ICLR.cc/2026/Conference — Submitted to ICLR 2026_

### Official Review · Reviewer_Wv3n · 2025-10-30

**Soundness:** 4
**Presentation:** 3
**Contribution:** 3
**Rating:** 6
**Confidence:** 4

**Summary:**

This paper proposes a technique to improve the ability to associate circuit designs with their respective specifications by training models to learn a joint representation over the two domains. The authors demonstrate the effectiveness of this technique by evaluating trained models on retrieval tasks, classification tasks, and generalizability.

**Strengths:**

I really like the problem this paper is tackling: it is unique yet important, and seemingly understudied especially in the age of large models. The hypothesis is sound and builds on well-established findings, and the results reflect the advantages of the proposed technique. They also test the generalizability of the trained models by splitting the formulas.

**Weaknesses:**

I see some weaknesses in the paper:
1. Need for a contrastive learning approach: The paper has not really justified the need of using contrastive learning as such. While it would not harm the performance, I would assume given a dataset of ~300K pairs, the models would be trained in a rather straightforward manner.
2. Choice of encoder: I also am not sure why CodeBert specifically was chosen -- while it is trained on code, I don't expect the code in pretraining being considerably in distribution with respect to LTL formulas. I wonder what would have happened if we just used a standard Bert model or RoBerta instead of something specialized like CodeBert.
3. Generalizability experiments: I would be more convinced by the generalizability if you had shown model performance on composed specifications (if that is even possible), and more importantly, a held-out dataset consisting of specifications that were designed by professionals/that exist in model-checking textbooks or references. I also don't get a sense of how aligned your generated specs are with industry-standard specifications, even if it is just 100 samples.

**Questions:**

Refer to weaknesses.

---

> ### Author Response · Authors · 2025-11-20
>
> We thank the reviewer for their effort and the recognition of the importance of the problem and our approach to it. We are thankful for the constructive feedback.
>
> **Need for a contrastive learning approach: The paper has not really justified the need of using contrastive learning as such..... I would assume ..., the models would be trained in a rather straightforward manner.**
>
> Our baseline experiment, called "Siamese-CNML," is a supervised training approach that demonstrates the need for our contrastive method. We noticed that this experiment is misleadingly named, which we believe is the reason behind this confusion. Siamese-CNML is our adaptation of SentenceBert [1], a seminal work on learning semantic representations of text in a supervised process. However, as shown in Tables 1 and 2, this supervised training is inferior to the contrastive models. What we observe is a model that quickly learns the training dataset but fails to generalize beyond learning the "pairings" of circuit and specification apparent in the training dataset. We will expand on this baseline approach in the paper and rename it to something more descriptive.
>
> **Choice of encoder: I also am not sure why CodeBert specifically was chosen -- while it is trained on code, I don't expect the code in pretraining being considerably in distribution with respect to LTL formulas. I wonder what would have happened if we just used a standard Bert model or RoBerta instead of something specialized like CodeBert.**
>
> The reviewer's intuition is correct in that the choice of encoder model is not critical to our method's success. However, we do observe a slight benefit of transfer learning from training on code to LTL and AIGER. We now provide loss curves for the training of BERT, RoBERTa, and CodeBert models in Appendix R2. RoBERTa achieves comparable results to CodeBert but takes longer to converge to the same loss. BERT does take significantly longer to converge and is more sensitive to hyperparameters. While our experiments show that our approach does not depend on CodeBert, its use is beneficial.
>
>
> **Generalizability experiments: I would be more convinced by the generalizability if you had shown model performance on composed specifications (if that is even possible)**
>
> We evaluate generalization using the following experiment: We split the formulas in our dataset into smaller/simpler formulas (called CMNL-split) and train a separate model (called CNML-simple) on only those specifications. We then demonstrate that this CNML-simple model continues to perform well in retrieval and model checking with the full specifications (Tables 1 and 2). The performance of the CNML-simple model on the complete (i.e., composed) formulas demonstrates that our approach can generalize to these unseen compositional structures of the full specifications, despite being trained on simpler formulas.
>
> We further clarify potential misunderstandings: All evaluations in our paper are performed on the same dataset of unseen, full specifications (one could say, composed). We recognize this may not have been sufficiently clear in the paper, and we will revise Section 6.3 accordingly.
>
> **and more importantly, a held-out dataset consisting of specifications that were designed by professionals/that exist in model-checking textbooks or references. I also don't get a sense of how aligned your generated specs are with industry-standard specifications, even if it is just 100 samples.**
>
> Our dataset is based on patterns collected from the literature and industry studies. This includes work on the AMBA bus controller, small robot controllers, specifications of device drivers, and elementary examples such as adders, bit shifters, counters, mutexes, and arbiters. We refer to [2] for details on these patterns and to [3] for details on how these patterns were used to create the training and held-out datasets we currently use.
>
> We hope our responses address the concerns raised by the reviewer and help to make our contribution clearer. We are open to providing additional clarifications or conducting further experiments.
>
>
> [1] Reimers, Nils, and Iryna Gurevych. "Sentence-bert: Sentence embeddings using siamese bert-networks." arXiv preprint arXiv:1908.10084 (2019).
>
> [2] Jacobs, Swen, et al. "The 4th reactive synthesis competition (SYNTCOMP 2017): Benchmarks, participants & results." arXiv preprint arXiv:1711.11439 (2017).
>
> [3] F. Schmitt, C. Hahn, M. N. Rabe, and B. Finkbeiner, "Neural Circuit Synthesis from Specification Patterns", Advances in Neural Information Processing Systems (2021)

---

> > ### Comment · Reviewer_Wv3n · 2025-11-27
> >
> > Thanks for the additional experiments. I am still unconvinced by the following point:
> > Even if your dataset was based on patterns collected from literature and industry studies, it is still synthetically generated, unless I am misunderstanding something. My point being that I don't have a guarantee that the synthetically generated dataset is representative of *real-world problems* unless they happen to specifically contain those real-world problems. Another way to convince me of this is if you can show me a couple of the samples in your dataset, quantify its complexity, and show me how closely they resemble actual industry problems and the model's performance on that problem.
> >
> > Other than that, thanks a lot for your clarifications!

---

> > > ### Author Response · Authors · 2025-12-03
> > >
> > > We are glad that the clarifications were helpful! We hope to resolve this last concern as well.
> > >
> > > Based on the comment of Reviewer wqSR, we supplied several qualitative examples of circuit-specification pairs from our dataset, which you can also find in Appendix R4. To give a better sense of the semantic expressiveness of these formulas, we added annotations indicating where they belong in the Manna-Pnueli hierarchy [1], which classifies the complexity of the specifications. Circuit complexity can be directly derived from the number of latches in the circuit, with each latch contributing exponentially to the size of the corresponding symbolic representation. We hope that these examples help illustrate the semantic complexity of our dataset.
> > >
> > > While transferability to an industrial setting is an important aspect of our approach, we view our work primarily as foundational. For example, LTL is the foundation for specification languages such as SystemVerilog Assertions (SVA) or Property Specification Language (PSL), which are widely used in industry [2]. We plan to explore a more industrial setting in future work, but consider this out of scope for now.
> > >
> > > Nevertheless, we performed a new experiment over specifications and circuits from one of the industry case studies that we base our patterns on. For this, we extracted 12 specifications from the AMBA case study in [3], a formalization of ARM's Advanced Microcontroller Bus Architecture (AMBA). For the evaluation of these samples, we constructed a small dataset for cross-modal evaluation using the same technique as reported in the submission. We report the performance of our approach and baselines on these datasets in Table 7 in Appendix R5.
> > >
> > > We hope that this provides insight into our dataset and that the new results on the AMBA samples lend confidence to the relevance of the data.
> > >
> > > [1] Manna, Zohar, and Amir Pnueli. "A hierarchy of temporal properties (invited paper, 1989)." Proceedings of the ninth annual ACM symposium on Principles of distributed computing. 1990.
> > >
> > > [2] Armoni, Roy, Dana Fisman, and Naiyong Jin. "SVA and PSL local variables-a practical approach." International Conference on Computer Aided Verification. Berlin, Heidelberg: Springer Berlin Heidelberg, 2013.
> > >
> > > [3] Godhal, Yashdeep, Krishnendu Chatterjee, and Thomas A. Henzinger. "Synthesis of AMBA AHB from formal specification: a case study." International Journal on Software Tools for Technology Transfer 15.5 (2013): 585-601.

---

### Official Review · Reviewer_wqSR · 2025-10-30

**Soundness:** 3
**Presentation:** 3
**Contribution:** 2
**Rating:** 6
**Confidence:** 3

**Summary:**

This paper introduces a representation-learning view of model checking via Contrastive Neural Model Checking (CNML), which jointly embeds LTL specifications and AIGER circuits using two CodeBERT encoders trained with a CLIP-style contrastive objective. A large synthetic dataset (cnml-base) is built, and a single-guarantee variant (cnml-split) is derived by formula splitting for a generalization study. The method is evaluated on intra-modal retrieval with N=100 and N=1000, against algorithmic (e.g., Bag-of-Keywords, WL kernel) and neural baselines, an cross-modal retrieval (neural baselines only). Finally, they fine-tune for model checking and show generalization from single- to multi-guarantee formulas.

**Strengths:**

* A novel contrastive approach to neural model checking (joint embeddings for LTL and circuits).
* A synthetic dataset created by first sampling LTL formulas and then synthesizing matching circuits.
* Broad retrieval evaluation (cross-modal and intra-modal).
* Mini-batch construction that avoids duplicates and reduces off-diagonal false negatives.
* Clear, well-organized writing with intuitive embedding analyses (e.g., cosine-similarity distributions and a heatmap).

**Weaknesses:**

* Mini-batch false negative analysis is limited
* Cross-modal retrieval lacks non-ML baselines (e.g. edit distance between paired LTL/AIGER string forms).
* Runtime benchmarks versus non-ML baselines are missing scalability and variance are not characterized.
* No qualitative results overall (true positives/false positives/near misses, error taxonomy, or interpretability visuals).
* Sequence pooling is under-specified (CLS/mean/max/attention-pool not clarified), and its impact is not quantified.
* Practical metric such as “recall after model checking (top-k)” is not reported alongside raw recall.
* Potential data overlap/near-duplicates in synthetic sets are not analyzed; de-duplication is unclear.

**Questions:**

Please address the weaknesses listed above.

Please explain the sequence pooling layer in more detail.

---

> ### Author Response · Authors · 2025-11-20
>
> We thank the reviewer for their detailed and thoughtful feedback.
>
> **Mini-batch false negative analysis is limited**
>
> We agree this deserves more detail. Our analysis is based on measuring false negatives by model checking all off-diagonal pairs in multiple sampled mini-batches. The results of this analysis are the basis of the false negative rate of ~4% reported in Section 5.2. We explored various approaches to minimize false negative noise before settling on the methods described in l261-264 in our paper. While the complete elimination of false negatives proved infeasible, contrastive learning methods demonstrate robustness to label noise [1].
>
> **Cross-modal retrieval lacks non-ML baselines (e.g. edit distance between paired LTL/AIGER string forms).**
>
> The core limitation preventing the use of text edit distance between LTL and AIGER is that the two languages employ completely different syntax and semantics (for their detailed definitions, see Appendix A and B).
>
> We conducted a brief experiment to verify that there is no correlation between edit distance and positive or negative pairings. In the table below, we measure the Normalized Levenshtein Distance (NLD) of sampled positive and negative pairs of circuits and specifications. The edit distances between positive pairs of circuits and specifications are not only huge, but there is also no distinction between positive and negative pairs.
>
> We are not aware of any other possible baseline methods for the cross-modal retrieval except the brute-force model checking of the whole batch.
>
> Metric | Positive Pairs | Negative Pairs |
> | -------- | -------- | -------- |
> | Mean  NLD | 0.8904 | 0.8902 |
> | Std NLD |  0.0382| 0.0381 |
>
>
> **Runtime benchmarks versus non-ML baselines are missing scalability and variance are not characterized.**
>
> We have added wall-clock time measurements for the experiments in Section 6.2 of Appendix R1, as well as in Tables 5 and 6. These tables show the time required for each method to generate similarity scores over all candidates and to rank them.
>
> **No qualitative results overall (true positives/false positives/near misses, error taxonomy, or interpretability visuals).**
>
> We now provide several examples of specification-circuit pairs in Appendix R4 with both the AIGER text encodings and circuit visualizations. In Appendix R3, we include rank distribution histograms and cumulative frequency distributions for the retrieval experiments, along with an analysis comparing the methods.
>
> If the reviewer could clarify which analysis or visuals could strengthen our work, we remain open to implementing them in the remaining discussion time.
>
> **Sequence pooling is under-specified (CLS/mean/max/attention-pool not clarified), and its impact is not quantified.**
>
> For sequence pooling, we follow the HuggingFace implementation, passing the CLS token through a dense layer and a tanh activation function. We experimented with various pooling strategies (CLS, MEAN, MAX) and found no significant performance difference between them. We will incorporate this elaboration into the revision of the paper and thank the reviewer for their feedback on this point.
>
> **Practical metric such as "recall after model checking (top-k)" is not reported alongside raw recall.**
>
> As every problem set in our retrieval experiments has exactly one satisfying match by construction (l354-357), the raw recall that we report *is* the "recall after model checking (top-k)". We thank the reviewer for their feedback, and we will clarify this in the revised paper.
>
> Furthermore, we are open to adding other practical or interesting metrics that the reviewer might suggest for the retrieval experiments.
>
> **Potential data overlap/near-duplicates in synthetic sets are not analyzed; de-duplication is unclear.**
>
> We perform de-duplication at the pair level, removing duplicate circuit-specification pairs that occur after data generation. We additionally verify that train/validation/test splits contain no contamination.
>
> However, we do intentionally allow individual circuits and specifications to appear in multiple pairs. For example, the same circuit may satisfy different specifications, resulting in multiple pairs in the dataset that share the same circuit. This design choice preserves dataset diversity and reflects the reality where a single circuit can satisfy multiple specifications (and vice versa).
>
> To quantify this: over the full CNML-base dataset (n=295,665), on average, a circuit appears in 10.82 pairs, and a specification appears in 1.34 pairs.
>
> We believe these comments address all major concerns raised by the reviewer and that they helped clarify our contributions. We remain open to providing additional clarifications or running further experiments.
>
> [1] Xue, Yihao, Kyle Whitecross, and Baharan Mirzasoleiman. "Investigating why contrastive learning benefits robustness against label noise." International Conference on Machine Learning. PMLR, 2022.

---

> > ### Author Response · Authors · 2025-12-03
> >
> > Taking up the request for additional non-ML baselines for the cross-modal retrieval task: While text-edit distance would not work, as explained in our previous answer, we wanted to provide more insight into the alternative - brute force symbolic model checking. In this baseline, every instance in the batch is model-checked symbolically against the query specification. While this method achieves perfect scores (the approach is sound), the extreme computational cost limits its application. As seen in Tables 5 and 6, listing the run times, this approach takes ~160 times longer than machine learning approaches, taking more than 5.5 hours to complete the benchmark, whereas the neural models complete it in ~2 minutes. We added these results to Tables 5 and 6 in the Appendix.

---

### Official Review · Reviewer_CFRC · 2025-11-09

**Soundness:** 1
**Presentation:** 1
**Contribution:** 1
**Rating:** 2
**Confidence:** 5

**Summary:**

This paper uses contrastive learning to learn joint embeddings of linear temporal logic (LTL) formulas representing specifications, and and-inverter graphs formulas representing systems to be checked. Both kinds of formulas are in raw ascii text, and pre-trained CodeBERT models are used as encoders, which are further fine-tuned on synthetic tasks. Experimental evaluations show that, compared to vanilla CodeBERT and  Sentence-BERT, contrastive learning with CodeBERT outperform both on two retrieval tasks (i.e., cross-modal retrieval and intra-modal retrieval). Furthermore, CodeBERT after contrastive learning outperforms the original CodeBERT for downstream fine-tuning task, binary classification on circuit-specification pairs.

**Strengths:**

- the targeted research problem, model checking, is of great importance in hardware verification
- decent background such as LTL, And-Inverter Graphs, model checking, and contrastive learning, are provided

**Weaknesses:**

- contrastive learning has been widely explored in similar tasks like code analysis and theorem proving; applying contrastive learning for LTL specifications is fairly incremental, especially given this work simply applies this standard idea to a synthetic dataset generated with existing tools.
- evaluation tasks like cross-modal retrieval and intra-modal retrieval are artificial, and there is no clear indication how and to what extent, these retrieval tasks really help to tackle the model checking challenge (e.g., state exploration issue explicitly highlighted in the introduction).
- the findings is somewhat well-expected, CodeBERT with some fine-tuning on the synthetic dataset shall outperform the original CodeBERT.
- the assume-guarantee format is essential for this work, however, there is no discussion (including appendix) about the specific syntax for assume and guarantee sub-formulas. One concern is that they may be biased in a limited category.

**Questions:**

It is surprising that the authors believe LTL is short for "Linear-Time Temporal Logic" (see background section), especially given that LTL is the focus of this work. What is the complete syntax for assumption and guarantee sub-formulas? Are they limited in some category, for instance, certain operators like "implies" is not allowed.

Why are remaining N^2-N pairs (implicitly) considered negative? Shouldn't validation checking be performed?

How do cross-modal retrieval and intra-modal retrieval help model checking? Are they purely hypothetic or used in any model checkers? To what extent, do these retrieval affect the performance of model checking algorithms?

Certain discussions of the introduction (line 58 - 65) do not make much sense; if LLMs are used for the paper writing, the authors shall explicitly acknowledge that.

---

> ### Author Response · Authors · 2025-11-20
>
> We appreciate the reviewer's acknowledgment of the problem's importance. We believe many of the reviewers' concerns stem from misreading or overlooking key sections of our submission rather than flaws in the work itself. We address the concerns outlined in the following text.
>
> **It is surprising that the authors believe LTL is short for "Linear-Time Temporal Logic"**
>
> Linear-Time Temporal Logic (LTL) was introduced by Pnueli [1] in 1977, and the name, including its abbreviation, was coined shortly thereafter. LTL is regarded as a foundational logic, with a substantial body of research based on it [2,3,4], including two Turing Awards (1996, 2007). We also expanded the acronym in l119 in the paper for any reader who may be unfamiliar with the term. We are not aware of any other meanings of the abbreviation "LTL" within the domain of computer science.
>
> **the findings is somewhat well-expected, CodeBERT with some fine-tuning on the synthetic dataset shall outperform the original CodeBERT.**
>
> The reviewer's comment, unfortunately, oversimplifies our work. Fine-tuning a language model is not the focus of this paper; it is merely one component of our architecture and should not be considered a contribution when isolated.
> In contrast to the reviewer's belief, simple fine-tuning CodeBert on this problem is not straightforward. In our early experiments, concatenating specifications and circuits to feed into a single fine-tuned encoder resulted in complete memorization of the training dataset (random chance test accuracy). The more sophisticated approach of SentenceBert [8] is included as a baseline in our experiments, referred to as Siamese-CNML. As these experiments show, even that approach is far inferior to our method.
>
> **Why are remaining N^2-N pairs (implicitly) considered negative? Shouldn't validation checking be performed?**
>
> Not explicitly checking the N^2-N pairs is a standard practice in CLIP-like methods [5,6] and is a core component of the self-supervised methods [7]. Our contrastive learning approach and evaluation show that we do not need to validate all pairs, and that this is an explicit advantage of our work. We refer back to l224-233 and l254-257 in the paper where we discuss the benefits of the self-supervised contrastive approach.
>
> **What is the complete syntax for assumption and guarantee sub-formulas?** and **the assume-guarantee format is essential for this work**
> The assume-guarantee format is *not* essential for this work; it is simply the way our formulas are formatted. The assume-guarantee format impacts neither the syntax nor the semantic expressiveness of LTL. All sub-formulas are unrestricted LTL-formulas. A complete definition of LTL syntax and semantics (and therefore also of the subformulas) is given in Appendix A. Every LTL formula can be trivially transformed into assume-guarantee format by interpreting it as a single guarantee. We will further clarify this in the revision of the paragraph.
>
> **How do cross-modal retrieval and intra-modal retrieval help model checking?**
>
> The purpose of this paper is to learn representations of LTL formulas and AIGER circuits through model checking. This intention is reflected in the title and the paper itself. We demonstrate the utility of the representations (and therefore our approach) by providing three downstream experiments: cross-modal and intra-modal retrieval (Section 6.2) and model checking itself (Section 6.3). We do not claim that cross-modal or intra-modal retrieval helps with model checking.
>
> **Certain discussions of the introduction (l58 - 65) do not make much sense; if LLMs are used for the paper writing, the authors shall explicitly acknowledge that.**
>
> LLMs were not used for writing the paper beyond the purposes of grammar checking and polishing of the writing, as noted in the submission checklist. In addition to that, we have now added a detailed statement about the use of LLMs to the end of our paper.
>
> **Contrastive learning has been widely explored in similar tasks like code analysis and theorem proving**
>
> Contrastive Learning is a well-known paradigm in many areas of machine learning. We explore this in our Related Work section, where we differentiate our work from the state of the field. While we refer to the successful application of contrastive learning to theorem proving as explicit motivation of our approach (l86-87), theorem proving and code analysis are fundamentally different problems from model checking. The plentiful related work is not a weakness, but shows the importance of the field.
>
> We believe we have addressed the concerns and technical questions posed by the reviewer and pointed to specific locations in our submission where the answers to these questions were already present.
>
> We hope our response has been helpful, and we respectfully request that the reviewer reconsider their assessment of our submission. We are happy to provide additional information.

---

> > ### Author Response · Authors · 2025-11-20
> > **References**
> >
> > References
> > ---------------------------
> > [1] Amir Pnueli. The temporal logic of programs. In 18th Annual Symposium on Foundations of Computer Science, Providence, Rhode Island, USA, 31 October - 1 November 1977, pp. 46–57. IEEE Computer Society, 1977
> >
> > [2] Clarke, Edmund M., et al., eds. Handbook of model checking. Vol. 10. Cham: Springer, 2018.
> >
> > [3] Baier, Christel, and Joost-Pieter Katoen. Principles of model checking. MIT press, 2008.
> >
> > [4] Huth, Michael, and Mark Ryan. Logic in Computer Science: Modelling and reasoning about systems. Cambridge university press, 2004.
> >
> > [5] Radford, Alec, et al. "Learning transferable visual models from natural language supervision." International conference on machine learning. PmLR, 2021.
> >
> > [6] Zhai, Xiaohua, et al. "Sigmoid loss for language image pre-training." Proceedings of the IEEE/CVF international conference on computer vision. 2023.
> >
> > [7] Balestriero, Randall, et al. "A cookbook of self-supervised learning." arXiv preprint arXiv:2304.12210 (2023).
> >
> > [8] Reimers, Nils, and Iryna Gurevych. "Sentence-bert: Sentence embeddings using siamese bert-networks." arXiv preprint arXiv:1908.10084 (2019).

---

### Author Response · Authors · 2025-12-03
**Rebuttal Summary**

During the discussion period, we provided clarification to the reviewers. Both Reviewer CFRC and Wv3n asked about the advantages of our approach over simple fine-tuning. We clarified that simple fine-tuning leads to the memorization of the dataset and that the baseline "Siamese-CNML" is a supervised fine-tuning baseline based on SentenceBert, which demonstrates the advantage of our approach. We will rename this baseline for clarity. To further reinforce our results, we expanded the size of the datasets used to test our retrieval experiments and updated the tables in Section 6.2.

We followed the suggestions from Reviewer wqSR and introduced additional analysis of our retrieval experiments, including comparative analysis between the methods and a visualization of ranks using CFD plots in Appendix R3. Although a text-edit-based baseline for cross-modal alignment is not achievable, we provided a comparison against a brute-force symbolic approach, which, while accurate, is extremely computationally expensive and impractical. We added wall-clock time and its variance for all methods in our retrieval experiments to further illustrate this gap.

Reviewer Wv3n and wqSR requested qualitative samples from our dataset, as well as clarification on its connection to realistic samples from literature and industry. We provided several instances from our dataset in Appendix R4 and classified them in the Manna-Pnueli hierarchy to visualize their complexity. Furthermore, we provided a new experiment based on the AMBA bus controller, which indicates the applicability of our approach to real-world instances.

Following a question from Reviewer Wv3n, we conducted a small ablation study comparing different variations of BERT architectures and demonstrated that our approach is agnostic to the choice of encoder. All models converge, with slight benefits to the stability and speed of convergence of CodeBERT, which motivated our adoption of it. We explained our approach to sequence pooling in more detail and reported our previous experience, which showed no practical difference between different pooling methods. We thank Reviewer wqSR for the question, and we will include this information in the manuscript.

We believe that these additions and clarifications have helped resolve reviewers' concerns.

We sincerely thank the ACs and Reviewers for their efforts during the discussion period.

---

### Meta-Review · Area_Chair_3Vwb · 2026-01-05

**Summary:**

This paper introduces CNML, the application of contrastive learning to learn a neural surrogate for model checking of circuits with LTL specifications. This neural surrogates is supposed to be an approximate model checker, but extremely faster. It can also provide some representations that can used for retrieval tasks.

The reviewers highlighted some positive aspects of the paper such as the need for fast surrogates for model checking. At the same time, some aspects have been criticised such as a) lack of motivation for the setting w.r.t. the need to learn the representations, which sounds synthetic, b) the presentation of the work, c) the use of certain architectures and baselines. Authors answered in a compelling way especially for c) and added some additional experiments reviewers asked (e.g., comparison in time with a symbolic model checker and some sanity checks about string distances with the specifications). At the same time, the first two points are not fully addressed imho.
Even if the paper got scores pointing to a borderline accept, I believe the paper leans more towards borderline reject, as I argue next.

Concerning a) I agree that the motivation to perform representation learning in the context of model checking is not backed up by a compelling real-world scenario. On the other hand, having a neural surrogate for classical model checking is indeed a very good motivation, but the authors do not discuss in depth or compare directly to other neural surrogates for model checking such as Giacobbe et al. (2024). This lack of this motivation or a clear discussion why other neural surrogates cannot be directly compared clearly undermines the paper.

Concerning presentation b) I agree with CFRC that the paper can be written in a clearer way, where the setting and the proposed loss is formalized much earlier (Figure 2 and the description appear only on Page 6 and it is still not clear at that point why one cannot learn a neural surrogate in other classical ways, e.g., with purely supervised learning).

Lastly concerning c) the authors claim to use a supervised baseline Siamese-CNML, but it is not clear how they trained it and why one cannot train a simple neural network to just predict if the model satisfies the given specifications in a supervised way. Furthermore, as highlighted by Wv3n the choice of the neural backbone are somehow arbitrary. Why not using a non-Bert LLM or any other neural net (e.g., an LSTM)? The authors added a mini ablation with RobertA and Bert but this does not answer properly the question.

I invite the authors to take another round to improve presentation, strengthen motivation and add compelling baselines.

**Reviewer Concerns:**

Reviewer CFRC essentially questions the motivation to learn representations for model checking. The authors push back saying they use model checking to learn representations instead. However they are still not answering the big question of why do we need these representations ultimately as their retrieval tasks seem ad-hoc and synthetic.
I hear the authors' complain about the misunderstanding the LTL acronym by the reviewer, but I don't believe the review is unethical.

wqSR instead gave a more positive score (6), and asked a number of technical questions about the experimental setting, pertaining point c), the lack of simple baselines. The authors did a good job answering mostly all of them. The only one I think are missing are ablations on sequence pooling (in the pdf they are not present), which link to the comment by reviewer Wv3n that the architecture chosen seem indeed arbitrary. This is the other point of concern left open.

**Reviewer Scores:**

CFRC would likely not raise their score, or at most raise it to 4.

wqSR would keep their score to 6, but could downgrade it to 4 during the discussion with the other reviewers.

Wv3n would probably lower their score to 4  during the discussion with the other reviewers.

---

### Decision · Program_Chairs · 2026-01-26

Reject